# The Complexity of Interferon Signaling in Host Defense against Protozoan Parasite Infection

**DOI:** 10.3390/pathogens12020319

**Published:** 2023-02-15

**Authors:** Silu Deng, Marion L. Graham, Xian-Ming Chen

**Affiliations:** 1Department of Microbial Pathogens and Immunity, Rush University Medical Center, Chicago, IL 60612, USA; 2Department of Medical Microbiology and Immunology, Creighton University School of Medicine, Omaha, NE 68178, USA

**Keywords:** protozoan parasites, host defense, interferon, pathogenesis, signaling pathway, crosstalk

## Abstract

Protozoan parasites, such as *Plasmodium*, *Leishmania*, *Toxoplasma*, *Cryptosporidium*, and *Trypanosoma*, are causative agents of health-threatening diseases in both humans and animals, leading to significant health risks and socioeconomic losses globally. The development of effective therapeutic and prevention strategies for protozoan-caused diseases requires a full understanding of the pathogenesis and protective events occurring in infected hosts. Interferons (IFNs) are a family of cytokines with diverse biological effects in host antimicrobial defense and disease pathogenesis, including protozoan parasite infection. Type II IFN (IFN-γ) has been widely recognized as the essential defense cytokine in intracellular protozoan parasite infection, whereas recent studies also revealed the production and distinct function of type I and III IFNs in host defense against these parasites. Decoding the complex network of the IFN family in host–parasite interaction is critical for exploring potential new therapeutic strategies against intracellular protozoan parasite infection. Here, we review the complex effects of IFNs on the host defense against intracellular protozoan parasites and the crosstalk between distinct types of IFN signaling during infections.

## 1. Introduction

Interferon (IFN) was first discovered in supernatant from virus-infected cell cultures [1] and was shown to interfere with viral replication. In subsequent studies on IFN for its antiviral function, three groups of IFNs were discovered based on their distinct receptors and functions: type I (IFN-α/β/ε/κ/ω/τ/δ/ζ), which signals through the IFNAR1/2 receptor; type II (IFN-γ), which utilizes the IFNGR1/2 receptor; and type III (IFN-λs), which engages the IFNLR1 and IL-10R2 receptors [2,3,4]. Type I IFNs, encoded by distinct genes, include 13 partially homologous IFN-α subtypes in humans (14 in mice), a single IFNβ and several poorly described single gene products (IFN-ɛ, IFN-τ, IFN-κ, IFN-ω, IFN-δ and IFN-ζ) [5]. Most cell types possess the ability to produce and respond to type I IFNs, with autocrine and paracrine effects, in response to microbial stimuli such as exogenous nucleic acids [6]. IFN-γ, as the sole member of type II IFN, can only be released by immune cells, mainly T lymphocytes and natural killer (NK) cells, in response to special cytokines, such as IL-12 and IL-18, or microbial antigens through the recognition of pattern recognition receptors (PPRs) [7,8,9,10]. Type III IFNs encompass four subtypes, including IFN-λ1, IFN-λ2, IFN-λ3 and IFN-λ4, which are encoded by distinct genes closely located on human chromosome 19. The former three IFN-λs (IFN-λ1, IFN-λ2 and IFN-λ3) were simultaneously discovered by two different groups in 2003 and found to exhibit structural association with IL-10 and functional similarity with type I IFNs [2,3]. IFN-λ4 was discovered later, in 2013, through the sequencing of RNA samples derived from primary human hepatocytes treated with synthetic dsRNA mimicking hepatitis C virus infection (HCV). IFN-λ4 was identified to modulate the cell response to HCV treatment through the genetic variations within IFN-λ4 [11]. The three types of IFNs signal through distinct heterodimeric receptors, but similarly trigger relevant gene expression via the Janus kinase–signal transducer and activator of the transcription (JAK–STAT) signaling pathway (Figure 1). the type I and III IFN signaling cascades initiate from the activation of receptor-associated JAK1 and tyrosine kinase 2 (TYK2), which phosphorylates the cytosolic transcription factors, STAT1 and STAT2. The phosphorylated STAT1/STAT2 heterodimer interacts with transcription factor IFN regulatory factor 9 (IRF9) to form a tripartite signaling complex termed IFN-stimulated gene factor 3 (ISGF3) [12]. In contrast, IFN-γ activates IFNGR1/2-associated JAK1 and JAK2, leading to the phosphorylation and subsequent homodimerization of STAT1, which is the functional transcription factor in the IFN-γ signaling cascade and known as IFN-γ-activated factor (GAF) [13]. ISGF3 and GAF translocate to the nucleus and bind to specific motif IFN-stimulated response elements (ISRE) and IFN-γ-activation sites (GAS), respectively, in the promoter region of IFN-stimulated genes (ISGs), leading to the transcription of related ISGs [14]. The receptor for type III IFNs (IFNLR1/IL-10R2) is selectively expressed in epithelial cells, hepatocytes, and several immune cell types, including B lymphocytes, neutrophils, DCs, and macrophages [15], while receptors for type I and II IFNs (respectively IFNAR1/2 and IFNGR1/2) are ubiquitously expressed on nearly all nucleated cells [16].

It has generally been considered that IFN-γ is essential for anti-bacterial and anti-parasite immunity, whereas type I and type III IFNs are mainly produced and participate in anti-viral immunity [17,18,19,20,21]. Recent advances support that signaling triggered by all three IFN types results in important and distinct outcomes, including antimicrobial activity and immunomodulatory potential, which vary with respect to ISG profiles [14]. It is highly possible that the signaling pathways induced by different types of IFNs interact with each other as they employ similar signaling molecules, such as JAK and STAT1, and alter distinct but overlapping ISG expression profiles, leading to the final balance of the host response. It becomes clear that the host type I IFNs could function as a double-edged sword where they provide early resistance against acute viral infections but are detrimental to the host during certain bacterial infections and chronic viral infections [6], whereas type III IFNs may act similarly to type I IFNs in the host defense against viruses. An important role of this type of IFN has been identified in mucosal immune defense against other microbial pathogens in the intestinal and respiratory tracts, which is consistent with the distinct distribution of its receptor [17,18,19]. In addition, more than one type of IFNs can be induced by the same pathogens in the same host [15,20,21,22], which obvious complicates the situation since different types of IFNs might cause distinct effects on the pathogen–host interactions. The advances in IFN research have been reviewed extensively and will provide new insights into our understanding of the complex signaling network of various types of IFNs in the host defense against protozoan parasite infections [6,13,15,16].

## 2. IFN-γ and Intracellular Protozoan Parasites

IFN-γ is an important cytokine in both innate and adaptive immune responses during intracellular parasite infections (Figure 2). Elevated levels of IFN-γ is detected in both experimental animals and human patients following intracellular protozoal infections [23]. An extensive number of studies support a protective role for IFN-γ against the infection of intracellular protozoan parasites, including *Plasmodium*, *Toxoplasma*, *Cryptosporidium*, *Trypanosoma*, *Leishmania*, while a few studies also indicated that IFN-γ contributes to the pathogenesis of parasite infection (Table 1).

### 2.1. IFN-γ Production in Protozoan Parasite Infection

IFN-γ production in intracellular protozoan-infected hosts is predominantly mediated by NK cells [24,25,26,27,28], and CD4+/CD8+ and Ag-specific T cells [29,30,31,32,33] in innate and adaptive immunity, respectively. Other types of immune cells have been reported to produce IFN-γ during protozoal infection (Figure 2); for example, natural killer T (NKT) cells were shown to secrete IFN-γ in the liver of *P. yeolii*-infected mice [34,35] and a significant proportion of γδ T cells and αβ T cells were reported to produce IFN-γ in the peripheral blood of *Plasmodium*-infected children [36,37,38]. CD11b+ CD45^low^ microglia and CD11b+ CD45^high^ blood-derived macrophages were identified as the major non-T, non-NK cells expressing IFN-γ in the brain of *T. gondii*-infected mice, whereas group 1 innate lymphoid cells (ILC1s) were identified to produce IFN-γ in the small intestine in response to the oral infection of *T. gondii* in addition to NK cells and T cells [39,40,41]. The production of IFN-γ by immune cells can be negatively regulated by anti-inflammatory Th2 cytokines such as IL-10 and IL-4 [42,43,44].

**Table 1 pathogens-12-00319-t001:** Effects of IFN-γ in host following intracellular protozoan parasite infections.

**Protective Effects**			
Parasite Species	Treatments and Findings	Effects	Ref.
*P. falciparum*	Recombinant IFN-γ in human hepatocyte cell culture	Hepatic schizont development ↓	[45]
	IFN-γ production in infected children	Occurrence of high-density and clinical episode of infection ↓	[37]
	High IFN-γ level in infected patients	Occurrence of cerebral malaria ↓	[46]
*P. berghei*	Recombinant IFN-γ in rats, mice and human hepatocyte cell culture	Hepatic schizont development ↓	[45]
	Monoclonal IFN-γ neutralizing antibody in mice	Parasitemia ↑	[47]
	Early IFN-γ production in infected mice	Occurrence of cerebral malaria ↓	[48]
*P. vivax*	Recombinant IFN-γ in chimpanzees	Parasitemia ↓	[45]
*P. Chabaudi*	Recombinant IFN-γ in mice.	Parasitemia ↓ Intraerythrocytic parasites ↓	[45]
	IFN-γ^−/−^ mice	Parasitemia ↑	[49]
*P. cynomolgi*	Recombinant IFN-γ in rhesus monkey	Hepatic schizont development ↓	[45]
*P. yoelii*	Recombinant IFN-γ in mice	Parasitemia ↓	[50]
	IFN-γR^−/−^ mice	Infection burden ↑	[51]
	IFN-γ^−/−^ mice	Parasitemia ↑	[49]
*T. b. brucei*	IFN-γ^−/−^ mice	Parasitemia ↑ Survival time ↓	[33]
*T. b. rhodesiense*	IFN-γ^−/−^ mice	Parasitemia ↑ Survival time ↓	[52]
*L. major*	IFN-γR^−/−^ mice	Larger and progressing lesions	[53]
*L. amazonesis*	IFN-γ^−/−^ mice	Devastating lesions in late infection stages	[54]
*T. gondii*	IFN-γ^−/−^ mice	Survival ↓ Infection burden ↑	[55]
*C. parvum*	Recombinant IFN-γ in intestinal enterocytes cell culture	Infection burden ↓	[56]
	IFN-γ^−/−^ mice	Survival ↓ Occurrence of chronic infection ↑	[57]
**Pathogenic Effects**			
Parasite Species	Treatments and Findings	Effects	Ref.
*P. berghei*	Late IFN-γ production in infected mice	Occurrence of cerebral malaria ↑	[48]
	Large amount of IFN-γ produced in infected mice	Susceptibility to cerebral malaria ↑	[58,59]
	IFN-γR^−/−^ mice	No cerebral malaria development	[60,61]
	IFN-γ^−/−^ mice	No cerebral malaria development	[62,63]
	Monoclonal IFN-γ neutralizing antibody in mice	Occurrence of cerebral malaria ↓ Survival time ↑	[64]
*P. yoelii,* *P. chabaudi,* *P. berghei*	Overproduction of IFN-γ	Development of Tfh and GC B cell response ↓ Humoral immunity ↓ Autoimmune anemia ↑	[65,66] [67,68] [69]
*T. congolense*	Overproducing IFN-γ in mice	Survival time ↓	[70]
	Reducing production of IFN-γ in mice	Survival time ↑	[71]
	Monoclonal IFN-γ neutralizing antibody in mice	Susceptibility ↓	[72]

### 2.2. IFN-γ in Host Defense against Protozoan Parasites

IFN-γ appears to be critical in controlling the infections of many intracellular parasites (Table 1). Exogenous IFN-γ was found to significantly diminish infections of *Plasmodium* in mice, rats, non-human primates, as well as in in vitro human hepatocytes, by inhibiting the parasite DNA replication during liver-stage development [45]. However, *Plasmodium* has also evolved a strategy to evade the host defense during liver-stage development by suppressing the expression of pro-inflammatory cytokines including IFN-γ in hepatic mononuclear cells [73]. *Plasmodium*-infected mice administrated with recombinant IFN-γ exhibited a suppressed blood-stage infection with the delayed onset of parasitemia, decreased levels of infected erythrocyte, and increased survival [50,74,75]. The positive effect of endogenous IFN-γ in the host defense against *Plasmodium* was determined from the increased susceptibility of rats treated with an anti-IFN-γ neutralizing antibody and in mice deficient in IFN-γ or the IFN-γ receptor [47,49,51]. Human studies also indicated a positive correlation between low IFN-γ production by live *Plasmodium*-stimulated peripheral blood mononuclear cells and the increased occurrence of symptomatic malaria as well as the risk of moderate-to-high-density *P. falciparum* reinfection [37]. Early IFN-γ production was shown to contribute to the protection against the development of murine cerebral malaria, the most severe neurological complication of *Plasmodium* infection, in *P. berghei*-infected mice and peripheral levels of IFN-γ were found to drop just before the onset of both human and murine cerebral malaria [46,48]. IFN-γ can be induced by malaria vaccines, as higher numbers of IFN-γ-producing T cells and increased IFN-γ level were detected in vaccine-treated subjects in several clinical trials [76,77,78]. Vaccination with chemically attenuated, asexual, blood-stage *Plasmodium falciparum* induces anti-parasitic cellular immune responses involving IFN-γ in *Plasmodium*-naïve volunteers [79]. Nevertheless, a subunit vaccine targeting *Plasmodium falciparum* circumsporozoite protein (CSP) activates the host immune responses dominated by parasite specific IgG antibody instead of IFN-γ [80]. Mice with a deficiency of IFN-γ or IFN-γ receptor have a higher susceptibility to *L. major* infection, accompanied by elevated Th2-type responses compared to the wild-type mice, but IFN-γ-deficient mice do not appear to succumb to *L. amazonensis* until 2 months post infection, suggesting that IFN-γ is induced at different stages of infection by diverse *Leishmania* species [53,54,81]. Similarly, IFN-γ or IFN-γ-receptor-deficient mice exhibited high susceptibility to infections by *T. b. rhodesiense*, *T. b. brucei*, and *T. gondii* [33,52,55,82]. While exogenous IFN-γ inhibits the development of *C. parvum* in cultured enterocytes without the need of immune effector cells, both IFN-γ-deficient and anti-IFN-γ-antibody-treated neonatal mice became more susceptible to *C. parvum* infection [56,57].

IFN-inducible cell-autonomous defense, including parasiticidal activity mediated by nitic oxide (NO), the disruption of parasitophorous vacuoles (PVs) related to IFN-inducible GTPase, the restriction of ion assimilation by natural resistance-associated macrophage protein 1 (NRAMP1), and the inhibition of nutrient acquisition by indoleamine 2,3-dioxygenases (IDOs), is critical for the restriction of parasite growth in infected cells and the elimination of the parasite inside the parasite-containing subcellular compartments [83] (Figure 2). Previous studies have underlined the role of nitric oxide synthase 2 (NOS2, iNOS)-NO in IFN-γ-mediated parasiticidal activity against intracellular protozoan parasites. IFN-γ-induced NO production showed the most evident parasiticidal activity against *T. cruzi* trypomastigotes and *L. major* amastigotes in IFN-γ-activated macrophages and *P. falciparum* as well as *P. yoelii* sporozoites in human and mouse hepatocytes, respectively [51,84,85,86]. Correspondingly, *Nos2*-deficient mice were highly susceptible to these pathogens [81,84,85]. NO production was reported to be induced by IFN-γ in hosts infected with *T. gondii*. Nevertheless, IFN-inducible NO might play a limited role and function at later time points in hosts infected with tachyzoites of the less virulent type II *T. gondii*, whereas it was essential in parasite control in virulent type I-*T. gondii*-strain-infected hosts [87]. *C. parvum*-infected mice had an increased level of NOS2 which was partially attributed to activated IFN-γ signaling [57]. A slightly longer infection period was observed in *C. parvum*-infected neonatal *Nos2*-deficient mice [88]. Nonetheless, several other studies with human enterocytes and mouse models indicated that the protective action of IFN-γ against *C. parvum* infection is independent of NO activity [56,89,90]. Thus, the precise mechanisms of NO-mediated antiprotozoal activity are still incompletely understood.

IFN-inducible immunity-related GTPases (IRGs) are a subfamily of IFN-γ-inducible GTPases characterized by a particular molecular weight between 21 to 47 kDa. They defend host cells against protozoa by targeting the PVs. IRGs can be divided into two groups—GKS-containing IRGs (IRGA, IRGB, IRGC or IRGD groups) and GMS-containing IRGs (IRGMs)—based on their canonical-lysine-containing (lysine, K) and non-canonical (methionine, M) G1 motifs (GxxG[K/M]S/T) within the conserved amino-terminal catalytic GTPase domain [91]. IRGM1, IRGM3 and IRGA6 enhance the IFN-γ-induced control of avirulent *T. gondii* strain in macrophages and astrocytes, which might account for the limited effect of NO in hosts infected with this parasite [92,93,94,95,96,97]. In host cells infected with avirulent *T. gondii*, IRGM proteins that act as guanine nucleotide dissociation inhibitors under an uninfected status are released from the endoplasmic reticulum which “turns on” the GKS-containing IRGs to target the PV [91]. IRGM1 also contributes to the elimination of *T. cruzi* infection in macrophages as IRGM1 KO macrophages displayed a defective intracellular killing of *T. cruzi* amastigotes [98]. A hierarchical model revealed the intrinsic order of IRGs when they are recruited to PV which indicated that IRGB6 and possibly IRGB10 bind to the vacuole before IRGA6, while IRGD is loaded last [99]. The recruitment of these molecules leads to vesiculation, membrane disruption, and sometimes necroptosis of the targeted PVs.

The other subfamily of IFN-γ-inducible GTPases, guanylate-binding proteins (GBPs), consists of a set of proteins with molecular weights between 65–73 kDa, comprising 7 and 11 members in humans and mice, respectively [100]. GBP genes are categorized into two clusters located on chromosome 3 and chromosome 5 in mice. Mice lacking GBPs on chromosome 3 (GBP^chr3^), including GBP1, GBP2, GBP3, GBP5 and GBP7, were highly susceptible to *T. gondii* infection due to the insufficient disruption of the PVs [101]. Moreover, macrophages lacking GBP^chr3^ showed defective loading of IRGB6 on the *T. gondii* PV membrane (PVM), suggesting that GBPs and IRGs coordinate with each other in PV targeting [101]. Compared with GBP^chr3^-deficient mice, mice deficient in GBP1 or GBP2 exhibited a milder decline of survival rates following *T. gondii* infection, indicating the complementary roles of each GBP on chromosome 3 in the host defense against parasite infection [102,103]. On the contrary, GBP1 was not recruited to *T. cruzi* compartments suggesting a protozoan specificity of GBP-mediated PV disruption [104]. The structural and biochemical cues of IFN-γ-inducible GTPase for targeting these molecules to the PV and whether membrane disruption is due to a direct effect of IRG activity, or a result of some intermediary molecules, remains unclear. Recent evidence suggests the C-terminal isoprenylation of GBP2 regulates the recruitment of GBP2 to the PVM by recognizing the ubiquitination on the PVM, which differentiates between the host membrane and the PVM [105]. E3 ligases such as TRAF6 and TRIM21 modulate ubiquitination of *T. gondii* PVM following IFN-γ treatment, whereas the dependent effect of these molecules on the IFN-γ-mediated elimination of *T. gondii* is controversial [106,107,108]. Interestingly, the distribution of GBPs in the host cytoplasm triggering the disruption of PVM is also regulated by ubiquitin-like autophagy proteins, such as autophagy-regulated gene 5 (ATG5) and GABARAP autophagy proteins, in an autophagy-independent fashion [109,110].

NRAMP1 is a highly hydrophobic integral membrane phosphoglycoprotein (~100 kD), expressed primarily in the late endosomal/early endosomal compartment of macrophages and polymorphonuclear leukocytes as a membrane transporter [111]. It has been shown to transport divalent ion cations, such as Mn^2+^, Fe^2+^ and Zn^2+^, to prevent the intracellular pathogens from these divalent metals essential for parasite survival [111,112]. NRAMP1 has been identified as an essential factor in the host defense against *L. donovani*, but the intrinsic mechanism remains unclear [113]. A previous study has revealed the correlation between cellular Fe^2+^ concentration and the IFN-γ-induced inhibition of *C. parvum* infection in intestinal enterocytes, but whether NRAMP1 is involved has not been investigated [56].

IDOs, IDO1 and IDO2, are both IFN-inducible, haem-containing oxidoreductases that degrade L-tryptophan to generate N-formylkynurenine (N-formyl-KYN) in the kynurenine pathway [114]. The removal of L-tryptophan restricted the growth of *T. gondii* in several IFN-γ-stimulated human cell lines including fibroblasts, lung epithelial cells, glioblastoma cells, retinal pigment epithelial cells, and macrophages [115,116,117,118,119,120,121]. Moreover, increased susceptibility to *T. gondii* occurred in mice with a double deficiency of IDO1 and IDO2 but not in IDO1-deficient mice, suggesting a significant role of both IDOs in the restriction of *T. gondii* infection in vivo [122]. IDOs may also control *T. cruzi* infection through the downstream kynurenine catabolites 3-hydroxykynurenine (3-HK) and 3-hydroxyanthranilic acid (3-HAA), which are likely to be harmful to *T. cruzi* amastigotes and trypomastigotes [123].

Additionally, IFN-γ could increase the expression of endothelial vascular cell adhesion molecule 1 (VCAM-1) to facilitate the recruitment of CD8+ T cells in the brain of mice chronically infected with *T. gondii* and enhance the cytotoxic potential of CD8+ T cells by inducing NO, which contributes to the host defense against parasites in the brain [40,124,125]. IFN-γ has been reported to modulate B-cell-mediated humoral immunity in *Plasmodium* infection via modulating the class-switching of antibody-producing B cells as IFN-γ-deficient mice produce less parasite-specific IgM, IgG3 and cytophilic IgG2a than wild-type mice [126].

### 2.3. IFN-γ in the Pathogenesis of Protozoan Infection

In contrast to the protective effect of IFN-γ, the response has also been reported to be involved in the pathogenesis of protozoan infection (Table 1). Although IFN-γ production as early as 24 h p.i. prevented the occurrence of experimental cerebral malaria in *Plasmodium*-infected mice, mice with late IFN-γ production at 3 to 4 days p.i. were found to develop severe experimental cerebral malaria [48]. IFN-γ mRNA accumulation was detected in mice susceptible to cerebral malaria [58]. The suppressed development of cerebral malaria was observed in *Plasmodium*-infected mice following the administration of anti-IFN-γ monoclonal antibody, treatment with IFN-γ-suppressive IL-10, inhibition of IFN-γ production, or deficiency of the IFN-γ receptor [59,60,62,64]. IFN-γ may mediate the development of experimental cerebral malaria through various mechanisms. IFN-γ, together with tumor necrosis factor (TNF) and lymphotoxin α, enhance the activation and apoptosis of the brain endothelium through the activation of endothelial cell and subsequently increased local binding of platelets [64,127,128]. IFN-γ is also necessary for the recruitment of CD8+ T cells in the brain by inducing the expression of canonical adhesion molecules, such as ICAM-1, CXCL9, and CXCL10. Accumulated CD8+ T cells mediate the immune responses against infected red blood cells sequestered in the brain and the lungs in susceptible mice, facilitating development of experimental CM [61,63]. The precise effect of IFN-γ in the development of cerebral malaria is still controversial. *T. congolense* highly susceptible BALB/c mice displayed significantly higher plasma IFN-γ levels compared to infected parasite-resistant C57BL/6 mice [72,129,130,131]. The IFN-γ-mediated accumulation and activation of erythrophagocytic myeloid cells led to acute anemia, liver injury, and a reduced survival time in *T. brucei* or *T. congolense*-infected BALB/c mice [70,71,132]. The overproduction of IFN-γ, induced by blocking IL-10R, shortened the survival time of both C57BL/6 and BALB/c mice following *T. congolense* infection [70]. *T. congolense*-susceptible BALB/c mice could switch to a relatively resistant-like phenotype by the neutralization of IFN-γ or by reducing the production of IFN-γ through the depletion of IL-12 during *T. congolense* infection [71,72]. Moreover, IFN-γ is also a crucial mediator in the humoral immunity that may exacerbate infection outcomes, leading to parasite-associated autoimmune disorders and chronic infection. *Plasmodium* DNA could induce autoreactive responses against erythrocytes by activating a population of B cells expressing CD11c and T-bet, in which IFN-γ acted as an essential factor, together with parasitic DNA to promote the expansion of autoreactive T-bet+ B cells, a major producer of autoantibodies promoting malarial anemia [69]. IFN-γ also contributed to the inhibition of T follicular helper cell differentiation during sever malaria infection, resulting in an impaired germ center B cell response and inefficient production of antibody-secreting plasma cells [65,66,67,68].

The precise role of IFN-γ in the host defense against protozoan parasite infection and in the pathogenesis of infection can be different during the infection of different pathogens, at different infectious stages, or in hosts with different intrinsic immune statues. Therefore, close attention to the alteration in IFN-γ levels and the IFN-γ-mediated immune response is necessary for timely adjustments of therapeutic strategies and predictions of prognosis of infection.

## 3. Type I IFNs and Intracellular Protozoan Parasites

Type I IFNs have been identified as critical immune regulators with diverse functions in infectious disease [6]. In contrast to supplying antiviral defenses for the host, these cytokines also exacerbate infection outcomes by suppressing the protective immune system or promoting proinflammatory events during infections, such as those caused by several viruses [133,134,135] and bacteria [136,137,138,139]. Recent studies support the double-edged functions of type I IFNs in the host defense against protozoan parasite infection.

### 3.1. Type I IFN Production in Protozoan Parasite Infection

The production of type I IFNs in intracellular protozoan infections involves multiple cell types through various signaling pathways (Figure 3). Hepatocytes produce type I IFNs during liver-stage *P. yoelii* and *P. berghei* infection in mice, but the molecular mechanisms involved in this induction remain controversial [34,140]. Leihl et al. demonstrated that both transcription factors interferon regulatory factor (IRF) 3 and IRF7 are critical mediators in a host type I IFN response and the specific cytosolic RNA sensor, melanoma differentiation-associated gene 5 (MDA5), recognized the pathogen-associated molecular pattern molecules (PAMPs) from *P. berghei* [140]. Miller et al. confirmed the involvement of IRF3 in host type I IFN production in a *P. yoelii* infection mouse model without the participation of IRF7 or all the known IRF3-related pattern recognition receptors (PPRs), including TLR3, TLR4, retinoic acid-inducible gene 1 (RIG-1) and MDA5 [34]. This inconsistency is probably due to the different *Plasmodium* species used in the infection models. During blood-stage *Plasmodium* infection, murine bone-marrow-derived dendritic cells (BMDCs) sensed *Plasmodium* RNA (*P. yoelii*, *P. berghei*, and *P. falciparum*) via the cytosolic RNA sensor MDA5 and subsequent adaptor protein mitochondrial antiviral signaling protein (MAVs) to activate the expression of type I IFN genes [140]. Plasmacytoid dendritic cells (pDCs; HLA-DR^+^ CD123^+^ CD304^+^) were identified as the major sources of type I IFNs in human blood-stage *P. falciparum* infection, whereas other peripheral blood mononuclear cells were also capable of producing type I IFNs [141]. Splenic red pulp macrophages generated significant quantities of type I IFNs in response to *P. chabaudi* infection in a TLR9-, MYD88-, and IRF7-dependent manner, in addition to plasmacytoid dendritic cells [142].

Murine macrophages and fibroblasts were reported to produce type I IFNs following infection by a few atypical *T. gondii* strains and the related parasite *Neospora caninum*, depending on endosomal TLRs, particularly TLR3, and cytoplasmic receptor RIG-1, respectively [143,144]. pDCs were able to produce low-levels of IFN-α in *T. gondii*-infected mice through the recognition of *T. gondii* profilin (TgPRF) by TLR12 [145]. Interestingly, several type I IFN-non-inducible *T. gondii* strains triggered type I IFN production in murine fibroblasts when they were heat-killed, but their co-infection with *Neospora* showed a strong inhibitory effect on type I IFN production suggesting *T. gondii* possesses type I IFN-inducing factors as well as the mechanism to suppress a host response to these factors [143,144]. A recent report demonstrated that IFN-β, a type I IFN, was produced by murine inflammatory monocytes in mesenteric lymph nodes following infection by *T. gondii*, whose induction is believed to be mediated through TLRs and Myd88 following the intake of the parasite [146].

As the major targeted cell type, macrophages were also found to produce type I IFNs in mice during infection with a non-metastatic Leishmania species *L. major* [147] and metastatic *L. guyanensis* [148], the latter of which was dependent on the TLR3-TRIF signaling pathway by recognizing parasite-derived factors, including dsRNA *Leishmania RNA virus 1* (LRV1). Metastatic but non-LRV1-contained *L. donovani* are capable of engaging type I IFN induction via TLR2, TLR3, TLR4, and downstream protein kinase R (PKR) in murine peritoneal macrophages [149]. These findings indicate that the macrophages are capable of producing type I IFNs in response to infection with different Leishmania species through the recognition of various parasite-derived factors via distinct signaling pathways.

The presence of type I IFNs upon *Trypanosoma* infection was first observed in the serum from mice injected with *T. cruzi* intraperitoneally [150,151], which was later proved to be IFN-α protein [152]. *T. cruzi*-infected murine fibroblasts were shown to secrete IFN-β, consistent with the discovery of an early type I IFN response in vivo at the site of intradermal infection [153,154]. IFN-β production was detected in murine macrophages and DCs following *T. cruzi* infection, which was dependent on TLR-associated adaptor protein myeloid differentiation primary response 88 (MyD88) and TIR-domain-containing adapter-inducing interferon-β (TRIF) [155].

The induction of type I IFNs in *Cryptosporidium*-infected hosts was revealed much later than 2009, showing the production of IFN-α/β in *C. parvum*-infected enterocytes and DCs, but the intrinsic mechanism of the induction has not been determined [156,157,158].

### 3.2. Type I IFN in Host Defense against Protozoan Parasites

Type I IFN signaling was shown to be beneficial to the host defense against the infection of several protozoan parasites (Table 2). *Ifnar*-deficient mice, or mice with *Ifnar* conditionally knocked out on hepatocytes, failed to control *P. yoelii* and *P. berghei* replication in the liver and resulted in a higher parasite burden in the liver and blood of infected animals [34,140]. Type I IFNs facilitated the recruitment of NKT cells which is essential for the elimination of *P. yoelii* in the liver of infected mice [34]. Using *Ifnar* KO mice, type I IFN signaling induced by *T. cruzi* was shown to be required for controlling parasite growth during the acute phase of infection by activating the production of NO in infected spleen cells [159]. A study involving *T. b. rhodesiense* reported a beneficial effect of IFN-I during the acute infection phase as *Ifnar* KO mice displayed a delayed control of parasite burden and succumbed to infection earlier than wild-type controls [160]. The production of protective NOS2-drived NO induced by type I IFN signaling was also observed in *L. major*-infected mice, while *L. major* appears to have evolved strategies to evade this immune response [147]. *L. major* infection induces the expression of a translational repressor, macrophage eukaryotic translation initiation factor 4E binding protein 1 (4E-BP1), and subsequently represses type I IFN production [161,162]. Moreover, *L. major* produces casein kinase 1 (L-CK1) to phosphorylate IFNAR1 on Ser535, leading to the degradation of IFNAR1 and subsequent attenuation of type I IFN signaling during infection [163]. *Toxoplasma* has also evolved strategies for limiting both the induction of type I IFNs and the ability of type I IFNs to activate STAT1-dependent transcription. Mice lacking the receptor for type I IFN-1 exhibited higher parasite loads and decreased survival following *T. gondii* infection [164]. Type I IFN expression was also induced by *C. parvum* infection in both human and murine intestinal enterocytes in vivo and exogenous type I IFN pretreatment inhibited parasite development in both cell types. The treatment of neonatal severe combined immunodeficiency (SCID) mice with anti-IFN-α/β-neutralizing antibody before infection significantly increased oocyst reproduction [156]. However, type I IFN signaling does not protect against *C. parvum* infection in IFN-γ KO immunocompromised mice. The type I IFN response reduced oocyst shedding and symptoms but not the intestinal parasite burden following *C. parvum* infection in IFN-γ KO mice. Moreover, the *C. parvum* infection burden was significantly lower in *Ifnar* KO mice compared to wild-type mice, suggesting that type I IFN signaling may dampen the host defense against *C. parvum* [158].

### 3.3. Type I IFNs in the Pathogenesis of Protozoan Infection

Similar to IFN-γ, Type I IFN signaling may also play a role in the pathogenesis of protozoan infection (Table 2). Robust type I IFN signaling in highly activated neutrophils was found to be associated with increased levels of serum alanine and aspartate aminotransferases, indicating liver damage [165]. Plasmodium infection in the brain of mice lacking type I IFN receptor was reported to be resistant to otherwise lethal cerebral malaria [166]. A study using conventional DC-specific Ifnar1-deficient mice and mixed BM chimeras showed that type I IFN signaling induced by *Plasmodium* infection negatively impacted cDC function, limiting the ability of cDCs, particularly splenic CD8^−^ cDCs, to prime IFN-γ–producing Th1 cells [167]. Type I IFNs suppressed the production of proinflammatory cytokines IL-17, IL-1β, IL-6, as well as IFN-γ, and promoted the development of immunosuppressive antigen-specific regulatory T cells (Tr1) to produce anti-proinflammatory cytokine IL-10 following *P. falciparum* infection [141]. One group, using Ifnar KO mice, showed that *Ifnar* KO mice could survive challenges with T. cruzi infection and were able to control parasite replication better than the wild-type mice [170]. Mice hyperresponsive for type I IFNs exhibited a significant defect in Th1 responses and IFN-γ production, suggesting that IFN-I plays a detrimental role in the early stages of disease. This corresponds to the finding that type I IFNs contribute to the downregulation of IFN-γ production and the loss of host resistance during chronic *T. b. rhodesiense* infection [160]. Type I IFNs were observed to suppress protective reactive oxygen species during human Leishmania infection which led to enhanced parasite burden [168]. *L. amazonensis-*, *L. major-*, or *L. braziliensis*-infected Ifnar KO mice developed attenuated cutaneous lesions and displayed a decreased parasite load, which is highly correlated to the robust recruitment of neutrophils, and promoted parasite-killing capability of neutrophils at early times post infection [169]. Hamsters challenged by a parasite strain harboring high Leishmania virus showed an increased production of IFN-β and higher susceptibility to Leishmania infection as well as more severe clinical symptoms [148]. Similar to that, in *P. falciparum* infection, type I IFNs induced by L. donovani amastigotes promoted the production of T-cell response suppressive IL-10, but through a distinct B-cell and endosomal-TLR dependent mechanism [171,172].

Current evidence suggests the roles of type I IFNs in host anti-protozoal defense and pathogenesis are controversial and depend on variation in the host species, parasite strains, parasite infection stage and location, and even parasite administration dose. The intrinsic mechanisms are mostly unclear. Future investigations need more precise methods to elucidate the exact spatial and temporal molecular activity inside different cell types from parasite-infected animals, and more studies are required to clarify the intrinsic mechanisms of immunomodulation and pathogenesis by type I IFNs.

## 4. Type III IFNs and Intracellular Protozoan Parasites

We know very little about the distinct role of type III IFNs in protozoal parasite infection as they are novel in this field. Type III IFNs were discovered as potent anti-viral cytokines similar to type I IFNs. Therefore, subsequent studies related to type III IFNs were mostly under the context of viral infections. However, recent studies revealed the production of type III IFNs in various bacterial and parasitic infections and assumed a potential immunomodulatory role in host defense against non-viral pathogens, even though current evidence is insufficient for a comprehensive conclusion [173].

A robust induction of type III IFN expression has been observed in patients infected with *P. falciparum* and in neonatal mice with *C. parvum* infection (Table 3). Transcriptome analysis identified IFN-λ as one of the top five differentially regulated cytokines in the blood of patients with febrile *P. falciparum* malaria. A subsequent study employing *P. yoelii*-infected *Ifnlr*-KO mice demonstrated that B-cell-intrinsic IFN-λ signals suppressed the acute antibody response, acute plasmablast response, and consequently impaired acute parasite clearance during a primary blood-stage malaria infection [174,175]. Clinical studies with *P. falciparum*-infected children in Kenya suggested a negative correlation between IFN-λ4 production and host immune protection against *P. falciparum*. Studies with both in vivo neonatal mice and in vitro intestinal epithelial cells showed that abundant IFN-λ3 was secreted in hosts in response to *C. parvum* infection [151,169]. Neutralization of IFN-λ3 exacerbated villus blunting and increased the fecal shedding of infectious *C. parvum* oocysts in neonatal mice, while prior supplementation of IFN-λ3 to intestinal epithelial cells reduced barrier disruption and enhanced cellular defense against *C. parvum* [176]. This *C. parvum*-induced type III IFN production has been attributed to the cell-intrinsic recognition mediated through TLR3 [158].

It is increasingly clear that although type I and III IFNs activate similar intracellular signaling cascades and gene expression in host cells, the two show distinct actions in infected hosts regarding the induction kinetics, tissue tropism, immunomodulatory effects, and even the ultimate impact on infection outcomes [177]. The unclear characteristics of type I and III IFN signaling in protozoan parasite infections prevent us from systematically understanding the commonalities and distinctions of their induction, regulation, and impacts during infection.

## 5. Crosstalk of IFNs

Protozoan parasites are able to induce the production of more than one type of IFN in the context of infection, in which different IFNs may act as similar protective effectors or play contrasting roles in the host defense against infection. *Plasmodium* liver-stage, as well as blood-stage, has been shown to induce both type I and II IFNs in the same host. A recent study using *Ifnar*^−/−^ and *Ifngr*^−/−^ mice determined that neither single type of IFNs established intact protection against *P. chabaudi* alone. The sum of the magnitude of the ISG response in the *Ifnar1*^−/−^ and *Ifngr1*^−/−^ animals was, on average, greater than the magnitude of the wild-type ISG response, suggesting the partial redundancy of the ISG response induced by these different IFNs [142]. However, revolutionary genetic studies revealed that some IFNs, including several subtypes of IFN-α as well as IFN-γ, have evolved under strong purifying selection to ensure their essential and nonredundant function in immunity to infection [178,179] (Figure 4). In some cases, type I IFNs act as an indirect enhancer of IFN-γ induction. For example, type I IFNs promoted the recruitment of NKT cells during liver-stage *P. yoelii* infection in mice to increase the production of IFN-γ [34]. Nevertheless, more evidence suggests that type I IFNs are potential negative regulators of protective IFN-γ signaling. Type I IFN signaling induced by *P. falciparum*, *P. berghei* or *T. b. rhodesiense* was reported to suppress Th1 cell development and subsequent IFN-γ production [141,160,167]. On the other hand, type I IFNs promote the production of anti-proinflammatory cytokine IL-10, an essential negative regulator of IFN-γ production in *P. falciparum* or *L. donovani* infections [141,172]. Fundamental questions can be asked based on these observations: why does the infected host produce different molecules playing a similar role “redundantly”? How do these distinct but closely related molecules interact with each other when induced in the same host? Does this interaction alter the infection outcome? The answers are complicated and remain open.

## 6. Conclusions and Speculations

Type II IFN (IFN-γ) mainly acts as a protective effector in the host defense against many intracellular protozoal infections and contributes to the pathogenesis of parasite infection in some cases. The precise roles of type I and III IFNs in host anti-parasitic defense remain unclear and require future investigations. The three types of IFNs activate signaling pathways which are distinguishable but share essential components, resulting in distinct but overlapping transcriptional outcomes. Current evidence highlights the various effects of different IFN responses on host anti-parasitic defense and implicates an indirect, regulatory, impact of type I IFNs on type II IFN production by immune cells. The partially overlapping ISG responses induced by the three types of IFN signaling in infected cells indicates potential direct intracellular interactions among various types of IFN signaling pathways, which currently lacks attention from the research field. Future studies should address the regulatory mechanisms of type I/III IFNs on IFN-γ signaling and the temporal kinetics of these interactions between different IFN signaling pathways in host cells following infection by protozoan parasites. This will facilitate monitoring disease progression and direct the administration of therapeutic strategies to protozoan infections. A better understanding of IFN crosstalk in the context of protozoan parasite infections will also provide new insights into mechanisms of the pathogenesis and host immune activities of other diseases involving multiple IFN responses, and contribute to the future development of IFN-related therapies.

## Figures and Tables

**Figure 1 pathogens-12-00319-f001:**
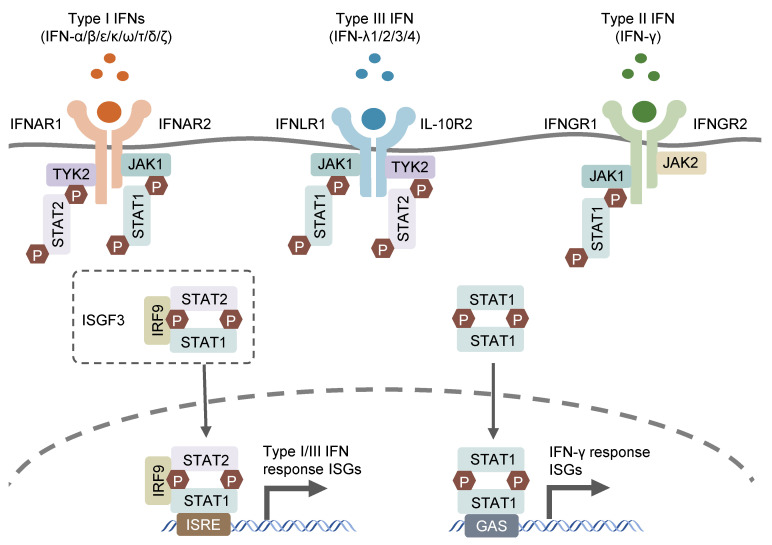
Canonic IFN signaling pathways. IFNs bind to their specific receptors on cells surface and activate the subsequent JAK-STAT signaling pathways to prompt gene expression of their related ISGs.

**Figure 2 pathogens-12-00319-f002:**
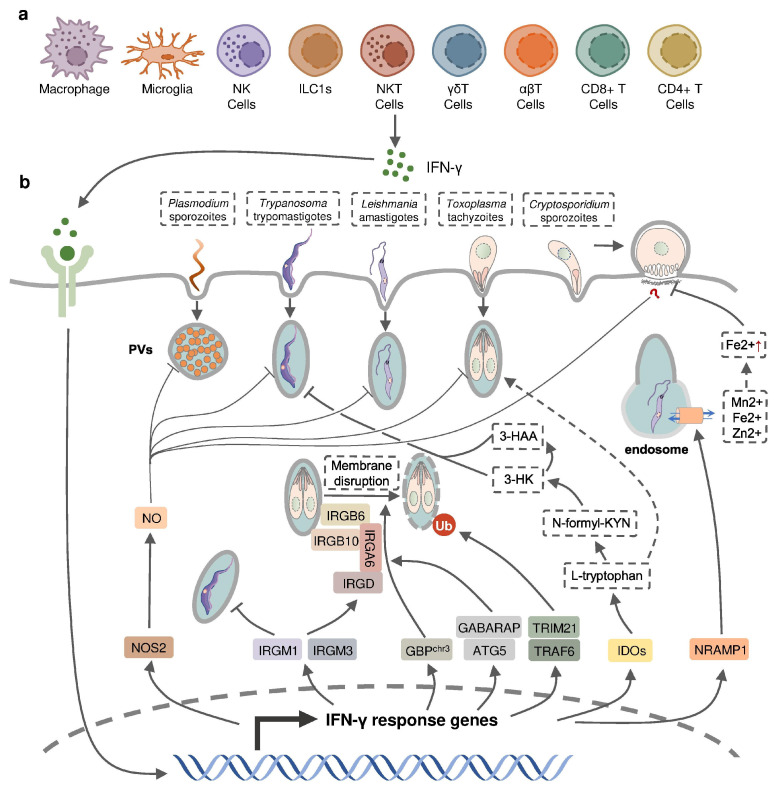
IFN-γ-inducible cell-autonomous defense against intracellular protozoan parasite infections. (**a**) IFN-γ can be produced by multiple cell types, including immune cells, during infection and (**b**) acts on infected host cells to eliminate intracellular parasite by nitric oxide (NO) production, the disruption of parasitophorous vacuoles (PVs) through IFN-inducible GTPase, the restriction of ion assimilation by NRAMP1, and the inhibition of nutrient acquisition by IDOs.

**Figure 3 pathogens-12-00319-f003:**
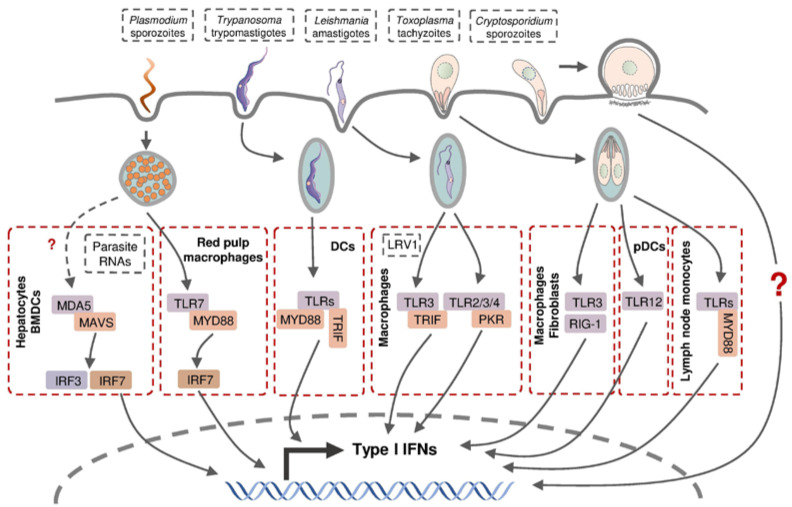
Type I IFN production in host cells following intracellular parasite infection. Parasite-derived PAMPs can be recognized by intracellular PRRs and thus activate subsequent transcription factors and stimulate the expression of various type I IFN genes.

**Figure 4 pathogens-12-00319-f004:**
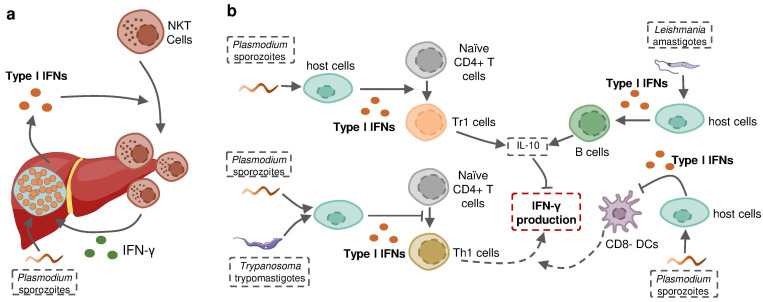
Type I IFN signaling induced by protozoan parasite infection regulates IFN-γ production during infection. (**a**) Liver stage Plasmodium-induced type I IFNs recruit IFN-γ-producing NKT cells to the liver and sequentially enhance IFN-γ production. (**b**) Protozoan-parasite-induced type I IFNs suppress host IFN-γ production by promoting the development of Tr1 cells and stimulating B cells to increase host expression of IL-10, inhibiting the differentiation of naïve CD4+T cells towards Th1 cells and restricting the activity of CD8- DCs to suppress the priming of IFN-γ-producing Th1 cells.

**Table 2 pathogens-12-00319-t002:** Effects of type I IFNs in hosts following intracellular protozoan parasite infection.

**Protective Effects**			
Parasite Species	Treatments and Findings	Effects	Ref.
*P. yoelii*	IFNαβR^−/−^ mice	Liver infection burden ↑ Parasitemia ↑	[34,140]
*L. major*	IFN-αβ neutralizing antibody in mice	Early parasite spreading ↑	[125]
*C. parvum*	Recombinant IFN-αβ in murine enterocyte cell culture	Parasite development ↓	[156]
	IFN-αβ neutralizing antibody in mice	Oocyst reproduction ↑ Gut epithelium infection burden ↑	[156]
*T. cruzi*	IFNαβR^−/−^ mice	Acute phase parasitemia ↑	[159]
*T. b. rhodesiense*	IFNAR1^−/−^ mice	Early parasitemia clearance ↓	[160]
**Pathogenic Effects**			
Parasite Species	Treatments and Findings	Effects	Ref.
*P. chabaudi*	IFNαβR^−/−^ mice	Liver damage ↓	[165]
*P. berghei*	IFNAR1^−/−^ mice	Survival ↑	[166]
	CD11c-Ifnar1^−/−^ mice	Neurological symptoms ↓ Survival ↑	[167]
*L. amazonensis*, *L. brazilliensis*	Recombinant IFN-β in human macrophage cell culture	Infection burden ↑	[168]
	IFNAR^−/−^ mice	Infection burden ↓	[169]
*C. parvum*	IFNAR1^−/−^ mice	Infection burden ↓	[158]
*T. b. rhodesiense*	IFN-αβ hypersensitive mice	Late phase parasite burden ↑	[160]
*T. cruzi*	IFNαβR^−/−^ mice	Survival ↑	[170]

**Table 3 pathogens-12-00319-t003:** Effects of type III IFNs in hosts following intracellular protozoan parasite infection.

**Protective Effects**			
Parasite Species	Treatments and Findings	Effects	Ref.
*C. parvum*	IFNLR1^−/−^ mice	Infection burden ↑	[158]
	IFN-λ neutralizing antibody in mice	Infection burden ↑ Oocyst reproduction ↑	[176]
	Recombinant IFN-λ in human intestinal epithelial cell culture	parasite development ↓	[176]
**Pathogenic Effects**			
Parasite Species	Treatments and Findings	Effects	Ref.
*P. yoelli*	IFNLR1^−/−^ mice	Infection burden ↓	[175]

## Data Availability

Not applicable.

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
