# Peer review of "The Complexity of Interferon Signaling in Host Defense against Protozoan Parasite Infection"

_pathogens, 2023, doi:10.3390/pathogens12020319_

Round 1

Reviewer 1 Report

The manuscript entitled "The Complexity of Interferon Signaling in Host Defense Against Protozoan Parasitic Infection". Title, abstract and overall rationale of work is well written and explain details about the IFN role in parasites. However, there are still some major concerns, which needs to be addressed.

1) The Abstract part written well and describe concise way.

2) In the keywords: Author must be add two more keywords that is important like signaling pathway and others

3) In the figure one author showing the role of type II IFN in parasites and rest of them showing the role in viral infection. However there are several published articles showed the effective role  of type I IFN Plasmodium (https://www.ncbi.nlm.nih.gov/pmc/articles/PMC6142290/ and  https://www.frontiersin.org/articles/10.3389/fcimb.2020.594621/full) for author reference and kindly read this review articles and redrawn the figure 1.

4) In this section (2.2. IFN-γ in host defense against protozoan parasites) author explain the role of IFN in plasmodium and author did not mention one important article that are showing the IFN-gamma role in Plasmodium (https://pubmed.ncbi.nlm.nih.gov/32181014/) author must be incorporate/cite this research article. However, this section written about the pathway and explain how to host defenses against the parasites. I recommend author to incorporate the figure here to show clear picture.

5) This section (2.3. IFN-γ in the pathogenesis of protozoan infection) author must describe role of NO, IL-10 and IL-4 cytokines because all these are interconnected to each other. This section written too much long and author need to reduce the incorporate the mechanism figure.

6) Author must be explain details about the IFN-gamma role in liver stage as well as blood stage infection. During vaccination how they behaves need to explain.

7) Conclusion section must be incorporate and this section should present at least in one 250-300 words paragraph and author must write future prospective and significance of this study.

8) There are some of punctuation and typographical errors throughout in the manuscript. kindly correct it.

9) Some references are too old and author must be revised for example reference no 24, 25, 27 and so on.

Reviewer 2 Report

This is an extensive review on IFN signaling and host defense against protozoa. The content appears to be accurate. My suggestions are mostly about grammatical, style and description accuracy. I suggest the authors writing the review in present tense, instead of past tense, it will read much better. Suggest paying attention to word choices for accuracy. Abbreviations are only needed to be defined once at first appearance. A few examples are included for the revision.

1.      Line 10: delete the word “other”, it should be “in both humans and animals”

2.      Line 11: change “socioeconomic loss” to “socioeconomic losses”

3.      Lines 15 and elsewhere: suggest change “protozoan parasitic infection” to “protozoan parasite infections”

4.      Line 36-37: IFNg can also be produced by Ag-specific t cells upon recall stimulation

5.      Line 45-46: clarify, not clear

6.      Lines 76-77: because you are talking about host responses to infections (or external stimuli), suggest changing “leading to the final balance and ultima outcome of infection.” To “leading to the final balance of the host response.”

7.      Lines 77-79: suggest modifying this sentence “It becomes clear that the production of type I IFNs could serve as a double-edged sword: type I IFNs provide early resistance against acute viral infections but are detrimental to the host during certain bacterial infections and chronic viral infections” to “It becomes clear that the type I IFNs can function as double-edged sword where they provide early resistance against acute viral infections but are detrimental to the host during certain bacterial infections and chronic viral infections”

8.      Line 86: modify “A detailed review of above advances has recently reviewed extensively [6,14,16,17]. These advances provide new insights into our understanding of the complex network of various type IFN signaling in host defense against protozoan parasitic infection.” to “The advances in IFN research have been reviewed extensively and will provide new insights into our understanding of the complex signaling network of various types of IFNs in host defense against protozoan parasitic infections [6,14,16,17].”

9.      Line 99: what do you mean by mastered?

10.   Line 119: change infection to infections.

11.   Line 120: change diminished to diminish; infection to infections.

12.   It is easier to say Plasmodium infection, and in similar situations.

13.    Line 124:  decreased levels

14.   Line 128: human studies or a human study

15.   Example of modification: “Mice with deficiency of IFN-γ or IFN-γ receptor showed higher susceptibility to L. major infection, accompanied by an expansion of Th2-type responses compared to the wild type mice, whereas IFN-γ-deficient mice didn’t show difference in susceptibility to L. amazonensis for the first 2 months of infection but show more severe disease afterward, which indicated that IFN-γ induction can be induced at different stages during infection of distinct Leishmania species [56–58]. Similarly, IFN-γ or IFN-γ receptor deficient mice exhibited high susceptibility to infection by T. b. rhodesiense, T. b. brucei, and T. gondii [37,59,60]. Exogenous IFN-γ inhibited the development of C. parvum in cultured enterocytes without involvement of other effector cells and both IFN-γ deficient and anti-IFN-γ antibody treated neonatal mice showed increased susceptibility to C. parvum infection [61,62].” To “Mice deficient in IFN-γ or IFN-γ receptor have higher susceptibility to L. major infection, accompanied by elevated Th2-type responses compared to the wild type mice, but IFN-γ-deficient mice do not appear to succumb to L. amazonensis until 2 months post infection, suggesting that IFN-γ is induced at different stages of infection by diverse Leishmania species [56–58]. Similarly, IFN-γ or IFN-γ receptor deficient mice exhibit high susceptibility to infections by T. b. rhodesiense, T. b. brucei, and T. gondii [37,59,60]. While exogenous IFN-γ inhibits development of C. parvum in culture without the need for immune effector cells, both IFN-γ deficient and anti-IFN-γ antibody-treated neonatal mice become susceptible to C. parvum [61,62].”

16.   Clarify this sentence: Recent studies support contrasting functions of type I IFNs in host defense against protozoan parasitic infection.

17.   Clarify this sentence: Hepatocytes were reported to induce type I IFN production during liver-stage P. yoelii and P. berghei infection in mice, but the molecular involved in this induction remains controversial

18.   Speculations section: it is very confusing please re-write some of the statements.

Round 2

Reviewer 1 Report

I have completed my evaluation of your manuscript and I found authors have addressed all the concerns raised in the previous version of the manuscript and the quality has improved after incorporating required modifications.

Author Response

Thank you so much for your quick evaluation of our manuscript.  We were glad to know that we have addressed all previous concerns and the quality of the manuscript has improved.  Much appreciated for your thoughtful comments and suggestions for our revision of the manuscript. 

Reviewer 2 Report

some minor suggestions:

1.      Line 108: delete “to produce”, duplicate

2.      Lines 144-151:

a.      Can you say: IFN-γ can be induced by malaria vaccines, as higher numbers of IFN-γ-producing T cells and increased IFN-γ levels were…

b.      What do you mean by this statement: Nevertheless, the predominance of the IFN-γ response in vaccine-induced host anti-malaria immunity remains unclear. Do you mean the predominant roles or levels of that cytokine? Suggest re-word or deleting this sentence as it does not add anything. The following 2 sentences are talking about IFNg and IgG, not sure how they might provide argument to the “lead sentence”.

c.      Suggest re-structuring to: Vaccination with chemically attenuated, asexual, blood-stage Plasmodium falciparum induces anti-parasitic cellular immune responses involving IFN-γ, but not parasite-specific IgG, in Plasmodium-naïve volunteers.

d.      Line 151, suggest deleting cellular immunity because IFNg alone is not sufficient to be the entire cell-mediated immunity

3.      Lines 338-340: this sentence is not clear. Suggested: A recent report demonstrated that IFNb, a type I IFN, was produced by murine inflammatory monocytes in mesenteric lymph nodes following infection by xxxx, whose induction is believed to be mediated through TLRs and Myd88 following intake of the parasite. What parasite???

4.      Line 442: typo, change to evidence

5.      Lines 501-519:

a.      This sentence is confusing: Current evidence highlights the various effects of the different IFN responses in host anti-parasitic defense and indicates an indirect regulatory impact of type I IFN on type II IFN signaling through modulating the production of IFN-γ by immune cells. Suggest modifying it to: Current evidence highlights various effects of different IFN responses on host anti-parasitic defense and implicates an indirect, regulatory, impact of type I IFN on type II IFN production by immune cells.

b.      Clarify this sentence: Future studies may demonstrate not only the immune cell-mediated but also the intra-infected cell regulatory mechanisms of type I/III IFN on IFN-γ signaling in infected host and the temporal kinetics of these interactions between different IFN signaling pathways. What is intra-infected cell?

c.      Suggest reword this sentence: A better understanding of IFN crosstalk in the context of protozoan parasite infections will also provide new insights on the pathogenesis and host immune activities of other diseases involving multiple IFN responses which will contribute to the future development of IFN-related therapies. To: A better understanding of IFN crosstalk in the context of protozoan parasite infections will also provide new insights into mechanisms of the pathogenesis and host immune activities of other diseases involving multiple IFN responses and contribute to the future development of IFN-related therapies.
